# Towards an operational European Drought Impacts Database (EDID)

Kerstin Stahl[1], Kathrin Szillat[1], Veit Blauhut[1], Monika Hlavsova[2], Lauro Rossi[3], Dario Masante[4], Andrea Toreti[4]

[1]Faculty of Environment and Natural Resources, University of Freiburg, 79089 Freiburg, Germany
[2] Global Change Research Institute CAS, Brno, Czechia
[3]International Center for Environmental Monitoring, CIMA Research Foundation, Savona, Italy
[4]European Commission Joint Research Centre, Ispra, Italy

*Correspondence to*: Kerstin Stahl (kerstin.stahl@hydrology.uni-freiburg.de)

**Abstract.** Drought impact information is essential to move from reactive management to a proactive approach. Data on drought
impacts provide regional insight into vulnerability and support robust risk assessment and sustainable adaptation strategies. Drought impact data are also essential to build and validate models for advanced impact forecasting, including AI enhanced tools. While there is increasing consensus on the operational use of specific physical drought hazard indices, to date there is no generally accepted convention on drought impact data collection and use. Based on experience and content of several regional research databases, the development of a European Drought Impact Database (EDID) explicitly aims for operational
application within the framework of the Copernicus European Drought Observatory. This article gives insight into the implementation of EDID, its structure and attributes, and provides an analysis of the content. Among the nine impacted systems, agriculture, public water supply and aquatic ecosystems contribute a majority of the impact records. Over the covered time period, impacts became more variable in the system they describe and recent years show some more extremely severe impacts according to a newly introduced severity score. Mapped at country scale, the impacts confirm previously identified
European sectorial impact hotspots. The work and product show that regional datasets can be integrated and add valuable information to an international European database. Public accessibility now provides the opportunity for update and improvement by mobilizing the European drought community. The service provided by EDID therefore has the potential to contribute to drought risk management and policy for a drought resilient society.

## 1 Introduction

Droughts are primarily measured by their associated hydro-meteorological conditions. They are often measured in terms of precipitation deficit, soil moisture anomalies, river discharge and groundwater anomalies. Nowadays, there is a general agreement on these variables describing the physical drought hazard as well as on indices used to estimate the onset, the intensity and the evolution of droughts. However, these approaches do not capture the complexity of droughts and do not directly provide data on observed impacts, nor on vulnerabilities that may contribute to drought risks (Hagenlocher et al., 2023;
Toreti et al., 2024). How to measure impacts and how data should be collected, processed and made available are less clear.

Impact data, i.e. systematic information that can be used for analysis, are essential to improve the understanding, characterization, and modelling of drought risks in different sectors and regions (e.g., Blauhut et al., 2016). Further, impact-oriented predictions as well as risk projections under different scenarios require impact data to develop and validate models. This is especially the case for recent models boosted by AI methods and tools (e.g. Sutanto et al., 2019; Stephan et al., 2023; Rossi et al. 2023; Shyrokaya et al., 2024, Bulut et al., 2025). So far, however, only a few operational early warning systems have made an effort to collect impact data and exploit their potential.

Defining, collecting, and sharing drought impact data aren't trivial tasks. Approaches, until now, have used somewhat different definitions; for instance the US Drought Impact Reporter (DIR), one of the longest established operational impact monitoring systems and data collection, defines drought impact as "an observable loss or change that occurred at a specific place and time because of drought" (Wilhite et al., 2007). Data are obtained from news media and observer networks and made available on a dashboard to provide local information as well as regional and national statistics and maps (NDMC, 2025). The European Drought Impact Report Inventory (EDII) defined drought impacts as "negative environmental, economic or social effects experienced under drought conditions" (Stahl et al., 2016). Impact data in EDII stem from a range of text-based sources that were gathered and categorized manually for several different research projects (e.g., Stephan et al., 2022). New dataset versions were therefore published on an irregular basis (e.g. most recently in Blauhut et al., 2022). A number of regional and national impact collections or monitoring approaches exist that target only a particular event (e.g., de Brito et al., 2020), a certain type of impact, or are for example used to monitor and publicly share water restrictions (e.g. propluvia.fr) or agricultural yield losses (intersucho.cz). A main difficulty in characterizing, detecting and attributing drought impacts is that they might develop with delay, even when hydro-meteorological indicators have already returned to normal conditions (Erian et al., 2021). Indirect or secondary impacts that are not a direct result of the water deficit often emerge at a distance from the drought-affected region and result from complex cascading effects and propagating shocks. Together with intangible impacts that are difficult to quantify, these types of impacts may be challenging to attribute to droughts. These differences and multiple options of defining and archiving drought impact information demand that any study or operational drought impact inventory carefully defines what a drought impact is for its purpose and choose an approach of collecting, selecting and sharing data specifically for that purpose (Stahl et al., 2024).

Droughts can affect many systems and sectors; consequently, a variety of impact categorization schemes exist that can be generalized into distinguishing three main broad domains: economic, environmental, and social. To understand the effects of drought on these general systems a more detailed delineation of the affected subsystems is needed and their definitions vary. Blauhut (2020) reviewed published drought risk analyses globally and found that out of 51 impact-informed studies, 33 looked at agriculture, 6 at reported impacts in general and the remaining studies were unique, i.e. targeting different systems such as hydropower and water resource availability. The majority of published studies focused on a single impacted system, with the studies on agriculture often even targeting specific crops (such as maize and wheat). Operational near real-time monitoring systems that actually use observations or quantitative impact data also use rather narrow system definitions with most of them focusing on agricultural conditions and agricultural losses (e.g. Trnka et al., 2020). The US DIR, being perhaps the only

operational multi-system drought impact system, uses 10 categories: agriculture, business and industry, energy, fire, plants and wildlife, relief, response and restrictions, society and public health, tourism and recreation and water supply and quality. The EDII used a more hierarchical system with 15 categories and altogether 105 subtypes, the large number of which reflects its research perspective. Across Europe, Naumann et al. (2021) found agriculture (crop and livestock), energy production, water supply, river navigation, and damage to buildings due to soil subsidence to matter most in economic terms. However, such a simplified approach neglects, e.g., the increasing importance of ecosystems and their feedback effects (via the services they provide) on economic and financial sectors. Collected reports in the EDII also found impacts on agriculture were most frequently reported across Europe. Other often-reported impacts relate to public water supply, aquatic ecology/fishery. Impacts on other sectors such as forestry, waterborne transportation, tourism etc. are more country- or region-specific and vary for different drought events (Stahl et al., 2016).

Other important considerations for operational drought impact monitoring are data availability in the targeted region(s) and feasible handling procedures to incorporate, update and provide added value for users. In the cited research applications, data incompleteness has been among the main issues. Therefore, research has recently focused more on automatic methods that widely search for impact reports in all available digital sources (e.g. de Brito et al. 2020). However, these approaches also have limitations, for instance related to language and media coverage in different countries. Regional and sector-specific databases for monitoring and decision-making have been in operational use already for some time (e.g. Trnka et al., 2020, Bartošová et al., 2021). The European Commission Joint Research Centre's (JRC) EDORA initiative (Maetens et al. 2024) was set out to build on those efforts and experiences and develop a method for the implementation of the first pan-European Drought Impact Database (EDID) serving operational purpose, within the framework of the Copernicus European Drought Observatory.

Here, we present the development of structure and content of EDID as a case for a transition from research on the impacts of natural hazards to an operational impact monitoring with the objective to make all data available to gain more knowledge about drought impacts. We specifically ask:

1. How can drought impact information be gathered and structured when moving from research to operation? Specifically, what are the challenges in harmonizing previous efforts into a new EDID that will also work operationally?

2. What is the empirical baseline of drought impacts in Europe? Specifically, which spatial and temporal patterns does EDID data reveal?

## 2 Data and methods

### 2.1 The service offered by EDID

While a design of a new operational database may draw on research experiences, it usually requires significant modifications to be operationally viable. The development of EDID included a user-friendly front end and an underlying database structure that efficiently integrates both previous data models and new future inputs. EDID is based on a geospatial database that contains

and interlinks different types of data, temporal, categorical, numeric, and text-based (Appendix 1). It is implemented to be accessed and used via a responsive web-mapping or through a more traditional tabular-based approach (Fig. 1). This functionality allows a user to interact with EDID via tabular and spatial dynamic systems, search for detailed drought impacts

as well as statistics to place an event into a broader context. An interface to process incoming data to the EDID format and allow to select and download data is implemented by the frontend web mapping interface. While the expert component of this web-service makes it possible to upload new entries, the public component allows downloading, visualizing and exploring all the content of EDID.

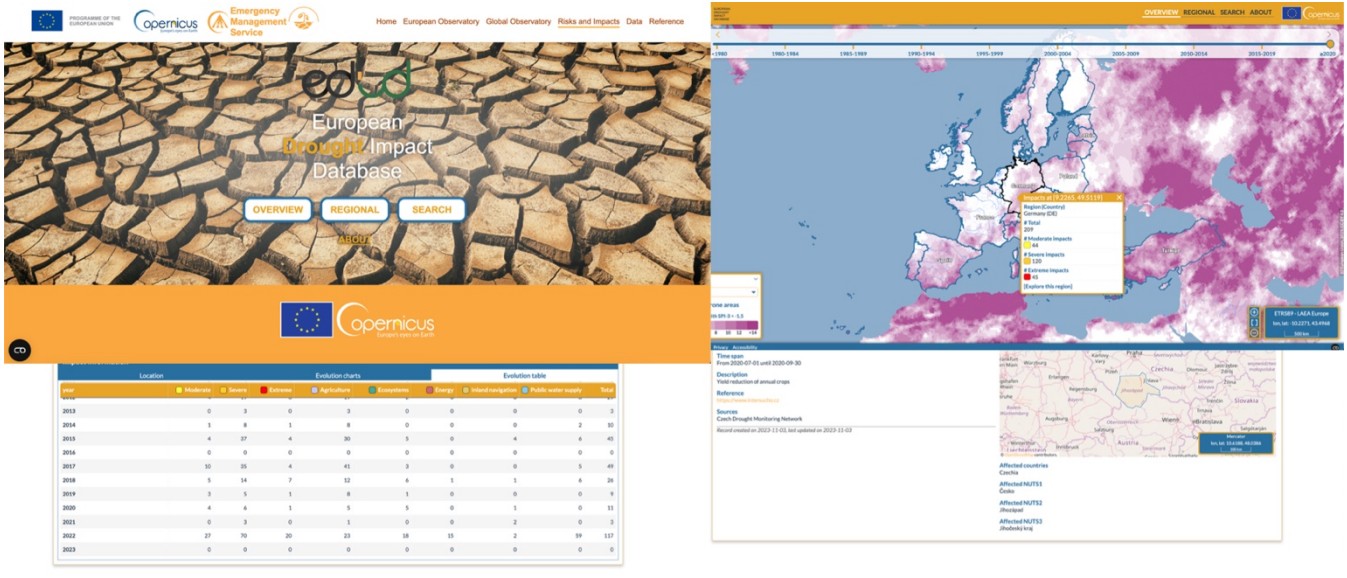

**Figure 1: Screenshots of the European Drought Impact Database (EDID) showing starting screen and tabular options of basic impact counts by severity and system (left) and mapping of the locations where these impacts occurred (right), among other options to explore impact records through the Copernicus Emergency web application (https://drought.emergency.copernicus.eu/tumbo/edid)**

The following definition of drought impacts was used for developing EDID: "Drought impacts are the direct and indirect, tangible and intangible negative effects of a drought hazard on environmental, economic and social systems. Depending on

the type of drought and the spatio-temporal occurrence of the hazard, also the type and severity of drought impacts can be specific to a region and system." The structural design of EDID also required a number of decisions that made previous datasets' wide options more parsimonious for operational handling while still keeping sufficient material to allow research and analysis in the future. Nine systems were identified as relevant in Europe according to the literature and the EDO user community of experts, allowing to distinguish impacts at the level of the affected system (e.g. agriculture-annual crops) while

avoiding too narrow or specific impact types (e.g. agriculture – strawberries). The systems are:

- Agriculture - annual crops: reduced productivity of annual crops (excluding impact on meadows)
- Agriculture - permanent crops: reduced productivity of permanent crops (excluding impact on meadows)
- Agriculture - livestock: reduced productivity of livestock farming

- Energy - hydropower: reduced hydropower production
- Energy - thermal: reduced energy production of thermal and nuclear power plants due to a lack of cooling water or the exceedance of temperature limits for return flows
- Inland navigation: impacts on river borne freight traffic with an impact ranging from a reduction of load to full closure of stream sections (excluding impacts on public transport and private water sports)
- Public water supply: local and regional problems and shortages in public water supply
- Ecosystems - terrestrial: all types of impacts on terrestrial environmental systems, including e.g. forests and grasslands
- Ecosystems - aquatic: all types of impacts on aquatic environmental systems

Each impact record then consists of:
- a georeferenced impact geometry as a core attribute for Web-GIS implementation,
- a main table of general attributes similar to all impacts regardless of the system, including the identifier of the affected system, the time of occurrence, a severity score, a summary description (translated to English), an information source (see Fig. 1 and Appendix 1), and
- optional additional attributes that might be system-specific ("ASSA"-values, see A1)

The georeferenced location where the impact occurred is important for any mapping and for the analysis of the relations with other geospatial information, e.g. hazard indices or vulnerability factors such as population. Many of the included datasets are based on administrative boundaries (from country to NUTS3 level), others used specific geographical markers such as rivers or lakes and/or specific coordinates. EDID can incorporate and store any geospatial geometry and internally keeps one geometry per impact record except when the event happens in regions that are not adjacent.

The timing of occurrence is an important attribute of an impact. For monitoring it allows to determine approximately a corresponding drought hazard event as defined by the EDO. The start and end point of drought impacts can be defined on annual, seasonal or monthly resolution in EDID. The minimum temporal reference required is the year of the impact occurrence; a more precise month or date is better. EDID introduces a 3-level drought impact severity classification. In brief, level 1 refers to warning, i.e. an expected impact with direct and local/individual influence; level 2 refers to direct and indirect impacts that are more widespread; level 3 refers to impacts related to enormous losses and cascading effects, irreversible deterioration, emergency actions over large areas. The levels, termed "moderate", "severe" and "extreme", were either assigned manually to each impact record, i.e. by expert judgement, or, in the case of data imported from EDII, by assignment rules for the 105 different 'impact subtypes' of EDII (see Szillat et al., 2025). For example, for the system agriculture-annual crops, reports about locally restricted expected losses on annual crop production were assigned the score of 1 (moderate); reports about documented more widespread reductions (<30%) or expected reductions > 30% and an actually documented reduction of annual crop production (> 30%) were given Score 2 (severe); documented "widespread, strong, very high" losses >30%, a

total failure of regional harvest and cascading consequences such as farmers going out of business or farm-related business consequences were given a score of 3 (extreme). As for other systems, the interpretation differs as the type of impacts does. The severity score should be considered as a starting point in need for refinement with more systematic data.

Besides those main impact descriptors, EDID provides a number of options for additional information (see Appendix 1). System-specific attribute tables that can be linked with an impact, for example, might be used to keep information from the original databases they were imported from. Alternatively, they can also be used to include impact information that may become available in the future, such as estimated economic loss values. Finally, there is the possibility to attach the original source of information to the impact, in terms of URL or file (e.g. pdf, word, etc.).

## 2.2 Drought impact data inclusion in EDID


Existing drought impact datasets helped to build the content of EDID. More datasets were initially considered, but during the development of EDID many were found to be either too weak with respect to their link to drought or too incomplete at the European scale; in other cases, they did not have all required attributes. The included impact information therefore mostly stems from text- or observer-based information used in previous data collections (Table 1). In addition, specific new searches
were conducted to fill gaps in space and time.

The main dataset used to develop EDID is the European Drought Impact report Inventory (EDII; Stahl et al. 2016). A range of reports and media outlets, as well as regular sectorial assessments and drought bulletins served as underlying text sources categorized into 15 categories and a total of 105 subtypes. The EDII 2.0 (Blauhut, et al. 2022) already integrated a number of extensions to the original dataset, mostly regional updates of EDII over the years, e.g. in Germany, the UK, and the Alps (Table
1). The EDII-Alps was created within the Alpine Drought Observatory project and extended the underlying sources by including information from operational monitoring such as the "propluvia" information system, an official ministry website on water restrictions in France and historic natural hazard databases such as the one in Austria (Stephan et al., 2020). Some regional databases also used EDII as a model to find and archive drought impact data. The datasets identified for the integration into EDID also include the DriDanube project and the Irish Drought impact database (Table 1). As part of the EDORA project,
a manual news media search helped update records for the year 2022, with a focus on Italy.

As the table shows, most of the past efforts to populate EDII were conducted in the framework of projects that focused on particular regions. A regional database that was also included into EDID is the observer-based sectorially-specific database by CzechIntersucho system (Trnka et al., 2020). This resource is based upon a monitoring network of volunteer agriculture reporters, evaluating impacts on production crops (both annual and permanent). Original data from Intersucho were aggregated
to the NUTS3 level and retrospectively assigned with missing mandatory attributes (e.g. severity).

**Table 1 - Drought impact data included in EDID**

| Original database | Underlying information and categorization | Area covered | Time period covered | Related projects/funding | Original No. of records (and no. transferred to EDID) | References |
|---|---|---|---|---|---|---|
| EDII 2.0 | Text-based categorization into 15 Impact categories and 105 Subtypes | Europe | 1970-2020 | | 17706 reported impacts (10790) | Blauhut et al. 2022 |
| EDII 1.0 | same, sources were different report types | Europe, with a focus on project regions | until 2015 | EU FP7 project: Drought R&SPI | | Stahl et al., 2016 |
| EDII 1.0 + | Text reports (various types) and survey (interview) data on hydropower and public water supply converted to categories and types | Additions and updates for Germany, and UK | until 2018 | Belmont Forum project DrIVER (DFG, NERC, NSF); Baden-Württemberg DRIeR project; unfunded Master theses | | Bachmair et.al 2017 (and further applications) |
| EDII Alps | Text-based additions from News Media and inclusion of coded information from regional monitoring system from France (propluvia.fr), Austria (hazard chronicle) and Switzerland (drought.ch) converted to EDII categories and types | Interreg's Alpine Space Region | until 2021 | EU Interreg Alpine Space Programme project ADO | | Stephan et al., 2021; www.ado.eurac.eu |
| EDII 2022 | Text-based additions from News Media focusing on the drought of 2022; manual search and categorizing into EDII system | several countries | 2022 | EDORA | | - |
| Czech Drought Monitor Intersucho | Observer-based questionnaires (farmers) coordinated and trained by CAS for Czech Drought Monitor: converted into EDID impact structure | Czech Republic and Slovak Republic | recent years | | (1254) | Trnka et al. (2020) |
| Irish drought impact database (IDID) | Text reports (various types), categorized into own Irish system (similarities to EDII) | Ireland | since 1800 | Irish National Project | 11000 (1412) | Jobbová et al. (2022) |
| DriDanube Drought impacts | Text report with a focus on news print and online media | Riparian countries of the lower Danube River | since 1981 | Interreg project: DriDanube | 926 (850) | Jakubínský, et al. (2019) |
| EDORA - automatic | Text-based online media sources found by semi-automatic web search in 24 EU languages & automatic translation | Europe | Identified gaps in the period 2010-2023 | EDORA | (1489) | |

EDID was also updated and gaps were filled in for major European drought events that were not well represented by the combined impact records. To accomplish this gap-filling, a semi-automatic procedure similar to the one employed by Hvlahova et al. (2025) was applied. In summary, it executes an automated online media content mining (coded in Python) using search terms including "drought" and other terms related ones to the nine EDID systems. For this purpose this one done in 24 European languages. The extracted article content was further processed with web-based translations to English and automated

text classification with the ChatGPT application programming interface. Finally, a supervised input to the database was performed.

## 2.3 Impact data analysis

Some details on the spatial-temporal structure of EDID must be clarified before subjecting the database content to spatial and temporal analyses. In EDID, an impact report that refers to multiple adjacent regions is saved as a single unique impact with
only one joint geospatial geometry. However, if a unique impact's description refers to multiple countries or NUTS regions and an analysis aims to provide counts or statistics at country or NUTS level, such an impact record will have to be counted in each country. Technically, to allow such a counting in each administrative unit, it may be useful to duplicate impacts for each spatial unit that is specifically mentioned in the source or even for all the units within a larger spatial unit, depending on the interest.

The first analysis focused on the frequency of the different drought impact attributes. It was based on the unique impacts. More specifically, it

- tracks the origin of the impact data and their classification into the systems. This is accomplished by visualizing the impact data flow from one categorization to another (with a "data flow" or "Sankey" diagram), aiming to facilitate understanding where the content comes from and how the content is distributed within EDID's structure;
- explores any changes over time in the frequencies of system-specific impacts and their severities, aiming to confirm major drought events by their impacts and to detect potential systematic changes in the attributes.

The second analysis targets country-scale impact record statistics. It therefore uses spatially distributed impact information data, i.e. counting impacts mentioned to have occurred in sub-units in the description as separate impacts. The results can thus
be compared to previously published analyses (e.g. Stahl et al., 2016). This spatial analysis across Europe specifically investigates

- each system's contribution to all impacts, hypothesizing that this metric represents each country's particular system (sectorial) vulnerability
- the severity score distribution of each system, hypothesizing that this identifies system-severity hotspots in Europe

## 3 Results

### 3.1 Drought impact data content: attribute distributions

The data that were included in EDID stem from different types of information sources (Figure 2). The relative majority – about 45 % - of all the impact records come from Media News. Online or printed news in the form of text, video and images make up this type. Within the EDORA project also a small fraction of government report sourced impacts provided impact
information. Due to the large share of government reports in EDII, this information type is the second most frequent type that

contributed to EDID. Around 28% of all the impact records originate from reports or press releases by a governmental organization including local authorities, national and supranational organizations, statistics offices. About 10% of EDID's initial entries stem from other inventories. Less frequent are impact records from scientific literature (7%), reports by NGOs (6%) and first-hand observations (3%) and all of these were included from EDII, which contributed 68% of the overall records. The other databases contributed between 5% and 9% and the EDORA project's new search added the remaining 10%. EDID does not contain impact records sourced from social media or via crowd-sourcing, nor does it contain impact records generated from models.

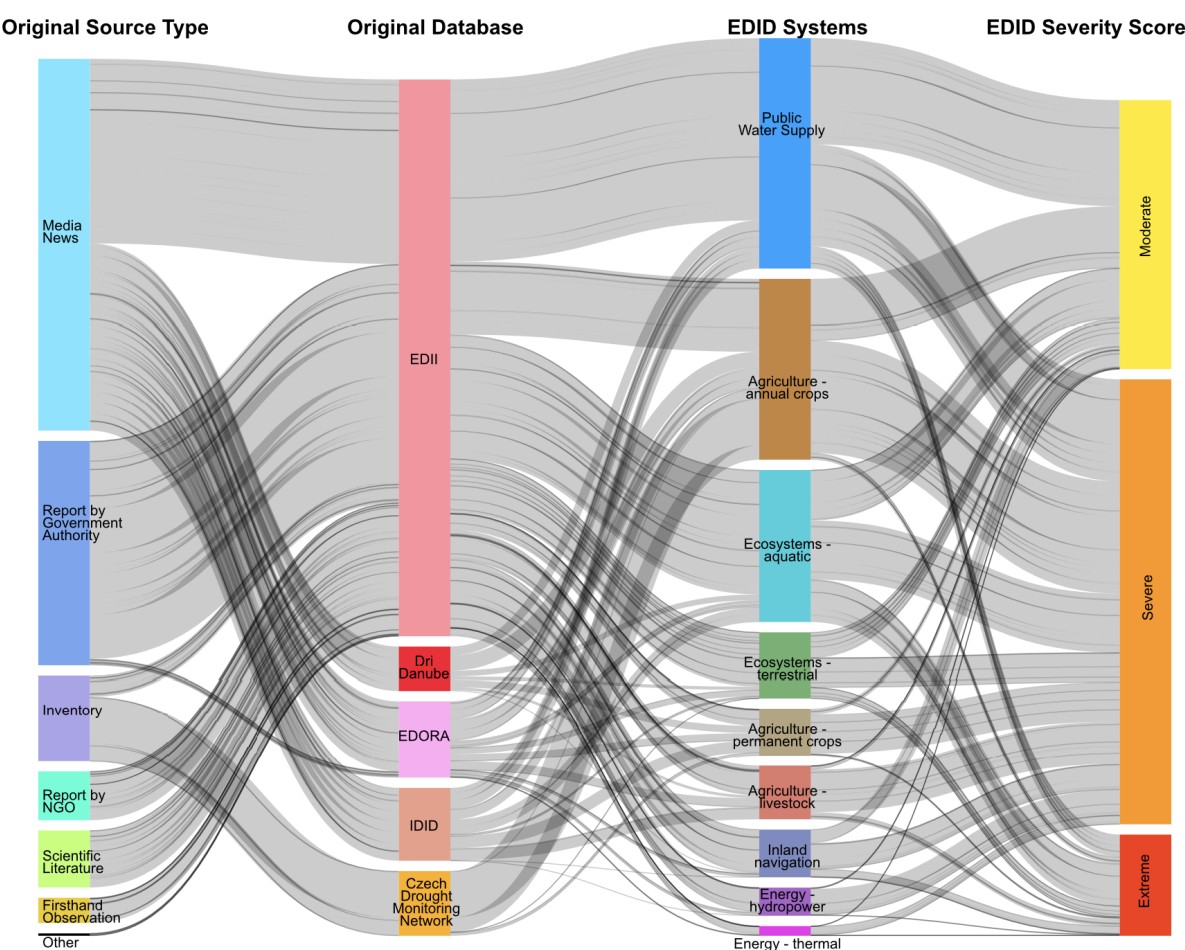

**Figure 2: Distribution of impacts in EDID over original information source types and original datasets (left columns) and their classification into EDID's nine systems and three severity scores (right columns). Numbers in % for the categories of each column are provided in the text.**

All impact records were classified into nine systems (Figure 2). The majority (28%) of all impact records belong to the system public water supply followed by ecosystems - aquatic (19%). The three agricultural systems together, however, make up the largest fraction (35%), with agriculture - annual crops (22%) being the largest of them. Agriculture - permanent crops and

agriculture - livestock comprise smaller fractions (around 6% each). Public water supply and ecosystem-aquatic inherited the majority of their impact records from EDII. Ecosystems - terrestrial (10%) also makes up a substantial part of the impacts. Inland navigation (6%) is less present and also the two energy-related systems, hydropower (3.5%) and thermal (1.2%). EDII, EDORA, IDID contributed to all systems, DriDanube to all but the energy system. The specialized Czech Drought Monitoring Network, which serves the agriculture and forestry sectors, contributed impact records to the three agricultural systems and to ecosystems-terrestrial, the system which also includes impacts on forests.

Overall, more than half of the impacts fall into the category "severe", about a third are classified as moderately severe and only slightly over 10% are classified as extreme. This distribution is broadly reflected also within each of the different systems with agriculture (permanent crops and livestock) and energy having a smaller fraction of moderate impacts and public water supply having a higher fraction of moderate impacts and ecosystems and inland navigation having a slightly higher fraction of extreme impacts (Figure S1).

Over the period 1970-2022 the number of impact records per year increased in EDID (Figure 3). Impact records became more frequent after the year 2003, which was a well-known severe drought year in central Europe. The time series shows notable drought years and multi-year periods of drought with over 200 impact records per year in 1976, in the late 1980s - early 1990s and 2003. The years from 2011 onwards have generally higher numbers of impact records with peaks in 2015 and 2018. The extremely severe records are more numerous in those years and their climatic anomalies have been described as main drought years (e.g. Orth et al. 2016, Toreti et al. 2019). Moderately severe records are more frequent in the past 10 years of the time series. In most of the recent peak impact years, EDID contains impacts in all nine systems. The trends and the severities differ somewhat among the systems (Figure S2).

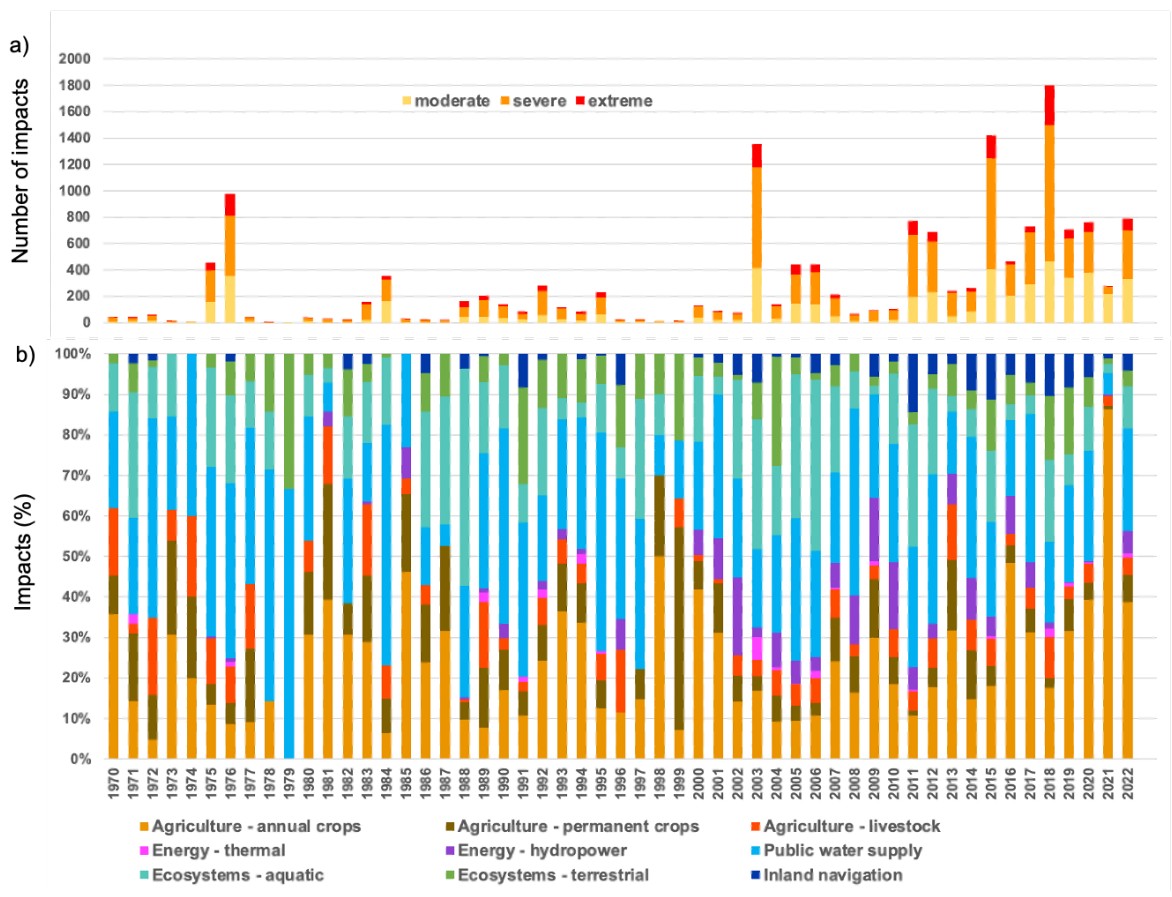

**Figure 3: Annual impact attribute distributions in the period 1970-2022 in terms of a) severity and b) fraction of systems affected.**

## 3.2 Spatial impact relevance and severity

The spatial distribution of drought impact records in EDID differs across Europe. The absolute numbers reflect the content of the transferred databases and their regional foci. EDID contains records for all nine systems in Croatia, Finland, France, Germany, Italy, the Netherlands, Poland, Romania, Sweden and Switzerland. For other countries not all the systems are represented, either due to the lack of data or the lack of impacts (e.g. inland transportation is not relevant everywhere). The consideration of the relative fraction of impact records in each of the nine systems helps remove this bias for those countries that have reports from a wide range of impacts (Figure 4).

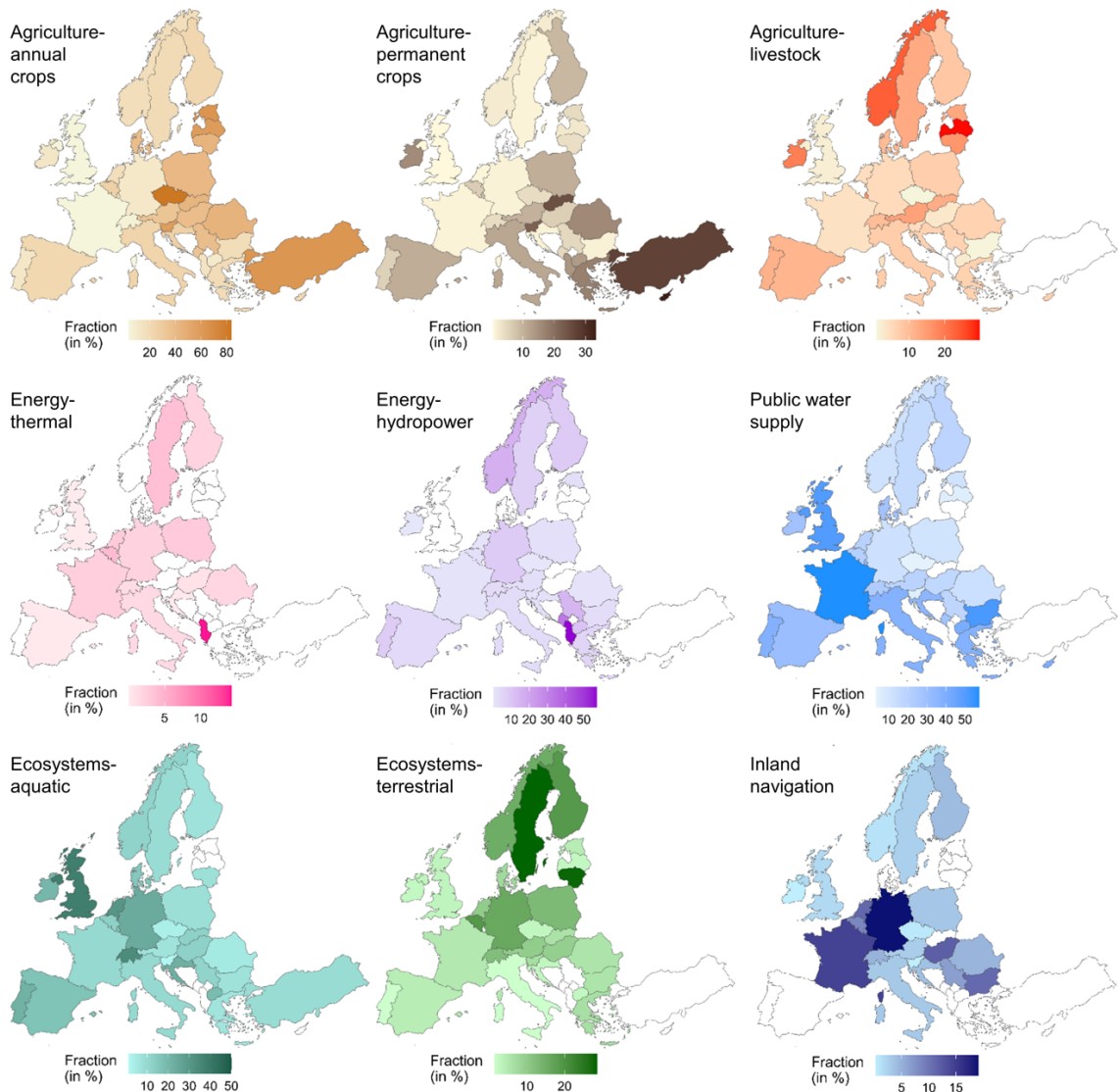

**Figure 4: Fraction (in %) of system-specific impacts by country**

Some regional patterns emerge from the maps in Figure 4. Impact records in the system agriculture - annual crops make up a large part overall (Figure 2), but regionally their relative fractions tend to be higher in southern and eastern Europe. The very high fraction in Czechia reflects the integration of the Czech national impact monitoring system. Records in the system agriculture - permanent crops make up a smaller part overall, but some countries show higher relative fractions, e.g. Czechia and some countries in southeastern Europe. Agriculture - livestock also contributes to few of the total records but some countries show higher fractions, e.g., Norway, Latvia, Ireland and Austria. Energy - thermal and energy - hydropower provide a small overall number of impact records. Nevertheless, country-specific fractions are high, e.g. in Albania, Sweden or Belgium (thermal) and in Albania, Montenegro, Serbia, Liechtenstein and Norway (hydropower). Public water supply is the system

with most impact records, but country-specific fractions vary geographically with higher relative values in France, the UK, Bulgaria and Cyprus. Ecosystems - aquatic is the system with the third most impact records. High proportions of impacts in this sector are found in the UK, Switzerland, the Netherlands and Germany. Ecosystems - terrestrial appears to contribute with high numbers of impact records in Sweden, Lithuania, Finland and Belgium. Impact records of inland navigation show a diverse spatial distribution, with higher proportions in Germany, France, Hungary, the Netherlands and Bulgaria.

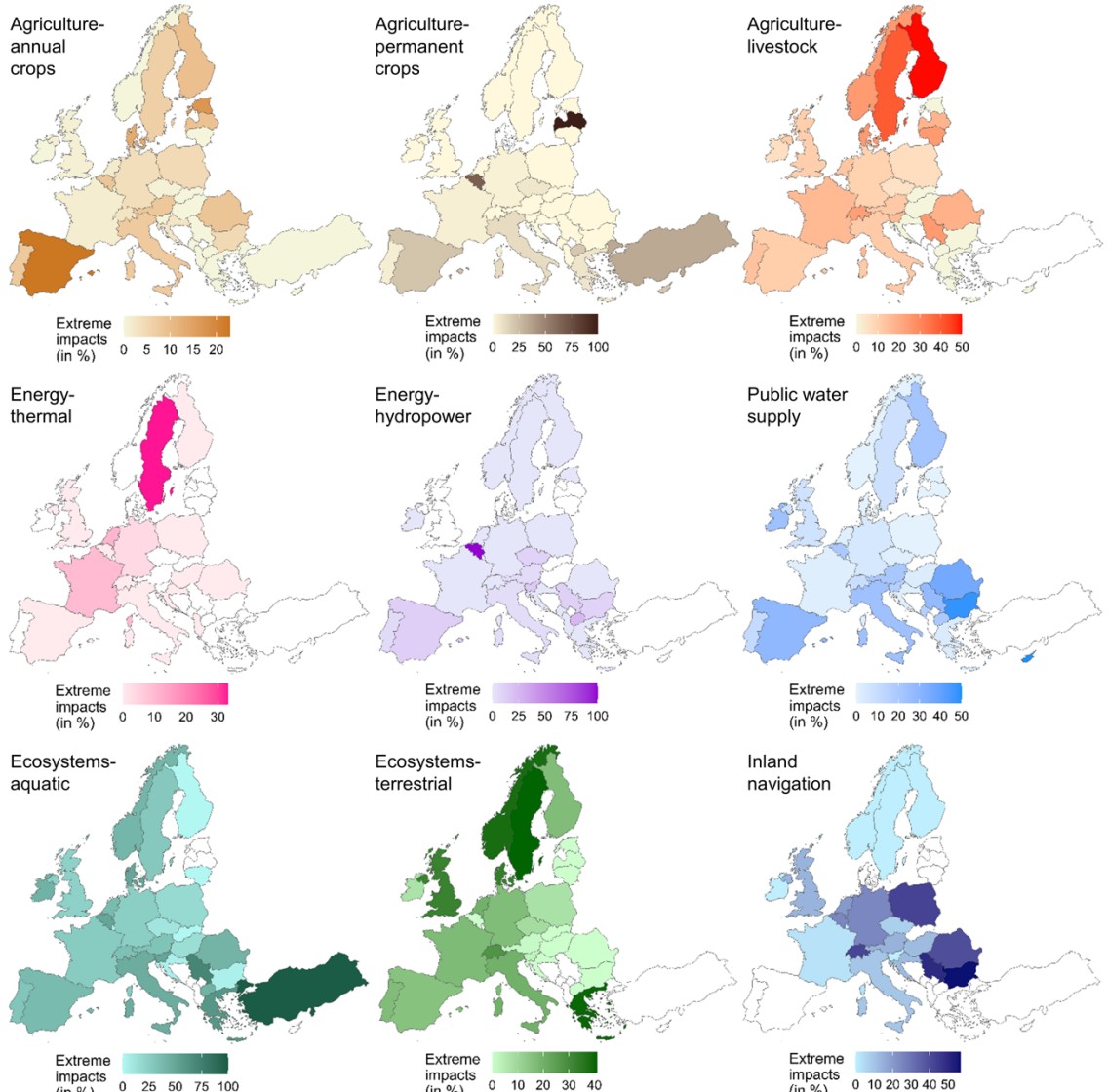

**Figure 5: Percentage of extreme impacts in each country and system; scaled to the total % of extreme impacts in the respective system.**

The severity scores of the impact records provide further information to understand and evaluate risks across Europe. The percentage of level 3 (extreme) impacts in each system and country suggests that a few countries and sectors were particularly

severely affected (Figure 5). Spain stands out with extreme impacts on agriculture-annual crops and permanent crops, while the Scandinavian countries appear to have had relatively extreme impacts in their livestock farming. Sweden, France and the Netherlands stand out with extreme severities in the thermal energy sector while Belgium's hydropower production appears to have been severely impacted, though the overall number of impacts is low. Concerning public water supply, the Mediterranean countries appear to have experienced relatively more extremely severe impacts. Extreme-severity impact

records on navigation suggest a hotspot in the lower Danube countries, but have also occurred in Poland, for example. In contrast to these regional patterns, in particular impacts that scored "severe" appear to have occurred more uniformly across Europe (Figure S3).

## 4 Discussion

### 4.1 A new baseline of drought impacts across Europe: spatio-temporal patterns

EDID provides a baseline for the assessment of drought risks across Europe from an impact perspective. With a large number of impact records consolidated from a range of previous datasets and complemented with new additions, EDID is the most comprehensive compilation of information across Europe since 1970. Overall, the distribution of impact records across the nine systems highlights agriculture as a major sector of concern. A similar regional dominance of impacts on agriculture of about a quarter to a third of all impact records was found previously in the EDII (Stahl et al., 2016) and in some of the other

datasets that are now integrated into EDID. An entirely independent text-based impact search by Sodoge et al. (2023) for Germany also found about a quarter of the impacts being related to agriculture. Hlavsova et al. (2025) also found this system to be one of the most reported in a global search for impact information. The results confirm that impact reports in the three agricultural systems taken together are the most frequent type of reported impact (Figure 3), but other impacts also matter. Public water supply is the largest single system in the database, confirming a known strong exposure of this sector across

Europe. Interpreting all the records on this system is more difficult, as there is a range of different types of impacts not strictly constrained to reports on 'drinking water', but to water supply for a range of purposes, restrictions, etc. Therefore, there is a need for further in-depth investigation to exploit all the opportunities and reveal all the patterns that may emerge from these data.

Impacts on aquatic ecosystems also have a strong representation in the database, confirming the importance noted in previous

studies together with the effects of the European Water Framework Directive (Stahl et al., 2016). The system ecosystems-terrestrial includes a large proportion of impacts on forestry and fewer impacts on a wide range of locally relevant other affected ecosystems. The drought response of trees often occurs with delay which might reduce awareness and reporting. It may be useful to combine the vegetation health indicators from remote sensing that are already used in many drought monitoring systems and the reported impacts, which can help the regional interpretation of these indices. The two energy systems and

navigation are much less represented in EDID. One reason might be that they are more restricted to the specific locations where they can occur, i.e. where special infrastructure exists that can be affected (e.g. water transportation routes, hydropower

plants). In addition, the impacts on people are less tangible and may be modified by market forces, which will likely affect the degree of awareness and limit reporting to regions where these impacts matter economically. Nonetheless the less represented systems are very important and future efforts might target them in a more focused way.

The spatial patterns of the impacts by country offer the possibility to identify differences in vulnerability and exposure to drought. These can be compared with other spatial analyses and with past studies that used other text-based archives, such as EDII almost a decade ago (Stahl, et al. 2016) and the European Drought Risk Atlas (based on statistical climate-risk models; Rossi et al., 2023). Maps as those in Figure 3 may still contain a high relative representation of impact records in a particular system in a particular country due to the selective import of system-specific data. For instance in Czechia, the inclusion of

agriculture and forest impacts from the Czech monitoring system is still visible in the relative fractions of the agriculture systems as well as ecosystem - terrestrial, which includes forestry. Another example for such remaining biases is the dominance of impacts in France in water supply, which is due to imported data from the propluvia.fr database. Also very low numbers of reported impacts and few collected records may influence the spatial pattern.

The maps in Figures 4 and 5 show that southern and eastern Europe are relatively more affected in agriculture - annual crops.

The severity score adds weight to this system. Southeastern Europe also appears to be impacted rather strongly in agriculture - permanent crops. High percentages of impacts in this system may reflect the importance of fruits and nuts/almonds there. Also Spain and Italy appear to be prone to extremely severe impacts in this system. Altogether, the weights confirm the exposure of Mediterranean countries to agricultural losses from drought, despite the diversity of socio-economic effects depending for example on rainfed or irrigated agriculture and on price effects (e.g. Espinosa-Tacón et al., 2022). Agriculture

- livestock's higher impact record proportions in Norway, Latvia, Ireland and Austria reflect a known importance of dairy and cattle in those countries (Diaz et al 2022; e.g. Jobbová et al., 2024). In the Scandinavian region, impacts in this system appear to have been extremely severe in the past. While agriculture is the main affected sector in southern and eastern Europe, energy - thermal does not show large differences across the continent, although extremely severe impacts appear to have occurred particularly in Sweden, France and Germany. A comparison with the European Drought Risk Atlas (Rossi et al. 2023) is

difficult as the analysis there considers only impact on nuclear power, but the lack of cooling water has wide ranging implications during drought. EDID confirms the importance of energy-thermal as a vulnerable system that would benefit from more research. Countries known for their hydropower installations such as Albania, Montenegro, Serbia, Liechtenstein and Norway show up in the maps, while the extreme severity consideration shifts focus to southern Europe. Impacts on public water supply are also present everywhere, while southern and in particular south-eastern Europe achieving the highest severity

scores. The impact pattern for water supply is very similar to the current risk estimated by the European Drought Risk Atlas. Impacts on aquatic ecosystems are also present throughout Europe, confirming the lack of a clear pattern mentioned in the European Drought Risk Atlas. Terrestrial ecosystems show higher relative occurrences in central and north-eastern Europe with extreme severity hotspots in Scandinavia and in Greece. While there is no clear pattern in the European Drought Risk Atlas, Sweden can be confirmed to be the most affected. Forestry is an important sector in Sweden that has suffered from

drought and follow up beetle infections (Aldea et al., 2024). Higher fractions of drought impacts on inland navigation in

Germany, France, Hungary, the Netherlands and Bulgaria reflect the major waterborne transportation routes on the rivers Rhine, Rhone and Danube. Poland and the lower Danube countries show severity hotspots. The European Drought Risk Atlas shows spatially elevated risks in some of those regions as well.

Those patterns and distributions of impacts have been changing over time. The annual impact records in EDID suggest that some areas of Europe experience the impacts of recurrent drought. Overall, we can observe an increase of records after the attention raising event 2003. The concurrent increase in reported drought impacts is also linked to the increasing availability of internet sources and online publications (incl. the onset of online news media) in that period. In addition, the gap-filling workflow employed within EDORA focused on the period from 2011-2022 and more diversity in the reported impact systems. Nevertheless, this shift to more and more diverse impacts also corresponds to an increasing frequency and intensity of meteorological drought and the role of temperature and evapotranspiration (Ionita and Nagavciu, 2021). A trend to more 'hot' or 'compound' (with heat) droughts (Hao et al. 2022) might be expected to alter the impact profile and compound droughts may increase overall water use and therefore pressure on water supplies.

### 4.2. Challenges and potentials of EDID: impact data for quantitative research and operation

A text-based impact database with pan-European scope was necessary to complement existing drought monitoring systems (such as EDO) and support a transition from hazard-focused to risk-oriented services. However, impact data such as the ones in EDID cannot be considered and used as biophysical monitoring data characterizing drought as a hazard. The mixed text and inventory-based sources that were chosen for EDID contain potentially subjective information on droughts that reflect regional relevance, awareness and culture. Objective interpretation with automatic, manual or hybrid methods to convert text descriptions and inventory data into EDIDs categorical and semi-quantitative impact information can reduce subjectivity, but temporal and spatial attribution can still hold uncertainty and be affected by some subjectiveness.

By transferring impacts from other databases, EDID benefitted from prior compilation of drought impact information. Some experiences are important for the future use of EDID and further developing of the content-building tools. Technically, transferring impact reports from other databases with different systems of categorization can be accomplished semi-automatically to some degree. For example Szillat et al. 2025 provides guidelines to transfer EDII-structured data to EDID-structured format. Nevertheless, some manual classification of highly diverse impacts of drought will always be necessary and the experience shows that it is best done by a person that has gained familiarity with the subject area.

EDID also benefitted from two other sources. First, the regional observer-based system from Czechia was used as a test case. Other systems, i.e. the US drought impact reporter also source impacts from observer networks for some impact types and while those may have specific foci, the experience shows that they might be a great source of information given that observers are sufficiently trained.

The use of automated web search approaches is useful, though it presents some challenges when applied at continental scale with respect to national efforts (e.g. de Brito et al. 2020, Sodoge et al., 2023). Accurate translation to and from many languages

is needed for search terms as well as for the harvested reports. Substantial tests with web-search tool workflows have shown that media services are powerful for short-term collection and impact. Automatic classifying of impact categories using large language models (LLMs) have revealed significant challenges, but are highly effective in summarizing query results, thereby reducing the time required for human processing.

Finally, it should always be kept in mind that the attention dedicated to drought in the media differs culturally. A consideration of cultural reasons why certain drought impacts are found worth reporting is largely lacking from data gathering, but will become important to consider when interpreting regional data. Demographics also control whether awareness propagates to an availability of reports and socioeconomic and political factors will affect the availability and quality of reports. More reliable in terms of consistency might be direct reporting by governmental authorities and trained experts. Overall, however, during the 18-month duration of the EDORA project about 1600 new drought impact records were added for many different European countries; compared to about the same number sources by the 3-year long DriDanube project, which relied on manual search by people in each of its partner countries. A conclusion is therefore that the web crawl translation and synthesis workflow can certainly speed up drought impact report collection and fill gaps, but that the process remains demanding in terms of time and human resources needed for reliable entries to the database.

The lack of shared criteria and standards, together with the fragmentation of drought information across actors and sectors, posed a challenge for systematic and comprehensive drought impact data collection. Beyond the collection of drought impacts, EDID may provide a model for recording drought impact data, suitable for both operational monitoring and research activity. In summary, the key to achieve this aim was a core component of basic information about the impact event (location, timing, sector, severity, etc.) for all records while providing flexibility for additional attributes and detail for specific applications. The flexibility also allows more sectors or additional attributes to be added without affecting the integrity of the database, and core components of records may be suitable to be transferred and adapted for different analyses or information systems. The inclusion of databases from further countries for example in Northern Africa or Turkey will be highly desirable, also considering that the physical drought indicators of the European Drought Observatory have been extended to that region. Hopefully such impact databases are being gathered in the respective countries for inclusion into EDID. This article aims to encourage such efforts and show the way to proceed.

Maintenance and updates are common challenges for any operational monitoring system. For EDID they are ensured by the Joint Research Centre (JRC) in the framework of the Copernicus European Drought Observatory leveraging also its role as co-chair of the EU Working Group on Water Scarcity and Drought. JRC has already developed a long-term plan based on the Europe Media Monitor and the use of large language models and the engagement of drought experts that can contribute via a dedicated component of EDID by adding new records.

**5 Conclusion**

The European Drought Impact Database EDID represents the most comprehensive collection of drought impact records from text-based sources available. Data are free and accessible via a user-friendly webservice integrated into the Copernicus European Drought Observatory (https://drought.emergency.copernicus.eu/). Despite some biases, EDID provides a new baseline for Europe. The records confirm European exposure to drought, with agriculture being the most affected sector in southern and eastern Europe. Public water supplies, aquatic ecosystems and energy production are affected all over Europe,

while inland navigation is impacted only in countries with economically important navigable rivers. EDID also reveals an interesting evolution in terms of increasingly reported drought impacts and the data may therefore help untangling the causal contributions of drought hazard, vulnerability and exposure.

  EDID has the potential to establish itself as a key resource for drought research and drought risk management. The update of the system will rely on automatic media monitoring tools and on community engagement. Indeed, the system is open to receive

new records from external qualified contributors and efforts to create such a community are ongoing.

**Acknowledgements**

  The entire EDORA Team contributed to the initial design of the database through discussion. The authors thank Simone Riolfo and Manuel Cavallaro, the main developers of the web application that serves as the interface with EDID, and Mirko D'Andrea

for his support with the semi-automatic gap-filling procedure. Ruth Stephan helped with the EDII-EDID transfer in the early phase. The contribution of all people who have provided impact information to the various data sources transferred into EDID are gratefully acknowledged. Jürgen Vogt and Lena Tallaksen initiated the roadmap to operation many years ago.

**Data availability**

  The searchable database is available at https://drought.emergency.copernicus.eu/tumbo/edid

**Funding sources**

  EDORA project (No 09200200.A092005/2021/862347/ENV.C.1—Lot 1) funded by the European

Commission Directorate General Environment (DG ENV). MH was supported by the Grant Agency of the Czech Republic, with project DynamicDrought no. 23-520 08056S.

**Author Contributions**

  Conceptualization: KS, VB, LR, MH, DM; data curation: MH, KSZ, LR; formal analysis: KS, KSZ; visualization: KSZ, KS,

LR; funding acquisition: KS, VB, LR, AT; writing (original draft preparation): KS; writing (review and editing): KS, KSZ, VB, MH, LR, DM, AT.

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

2007

## Appendix 1 EDID data model

Figure A1 provides an impression of the complexity of the data model of the EDID database and aspects in which it goes
beyond the mere archiving of the impact information. Operationally important components are for example, the user roles, the
archiving of any additional background material, or the connected hazard event that stems from other applications in EDO.

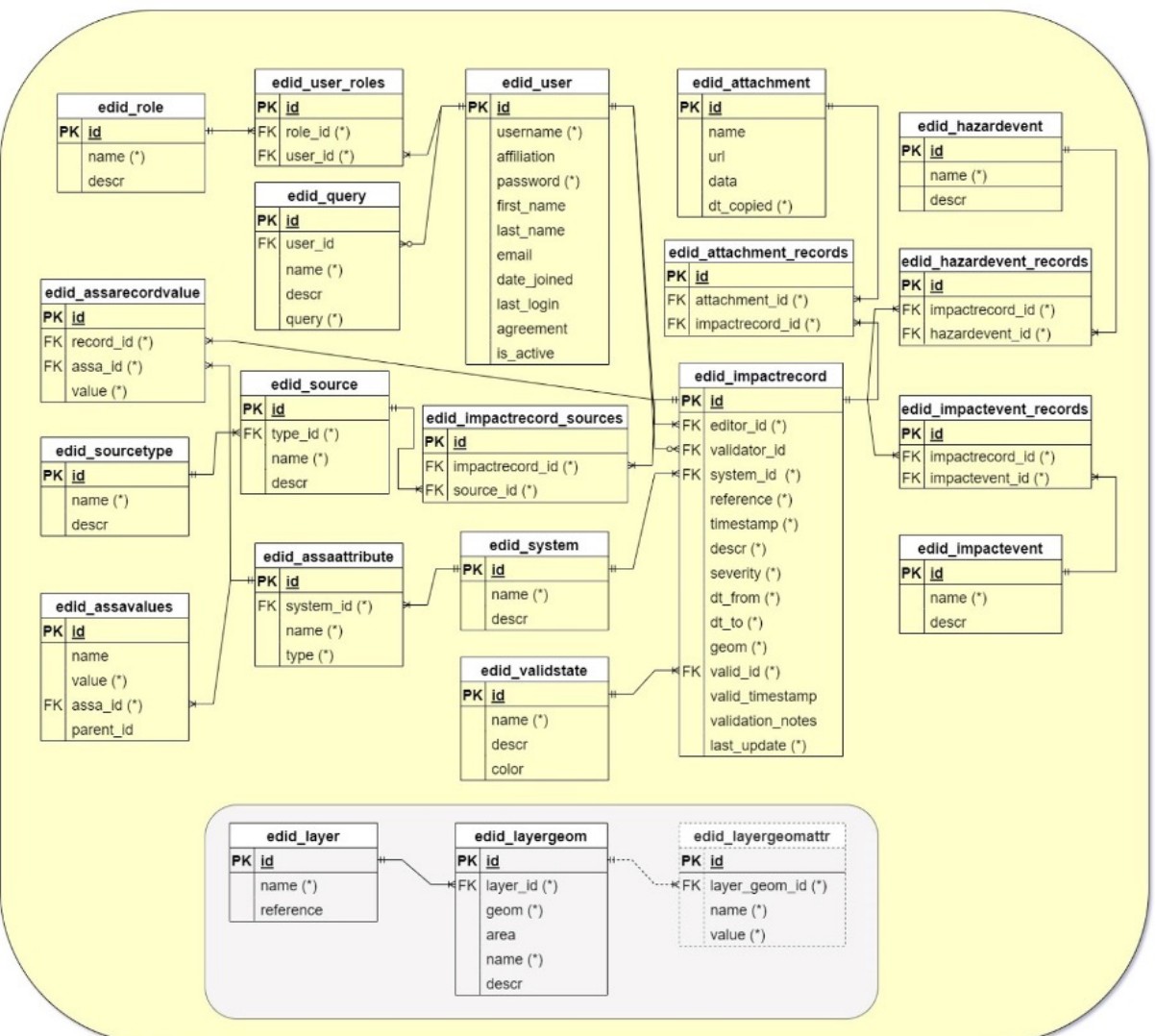

Figure A1 The EDID scheme. The following symbology is adopted: PK=primary key, FK=foreign key, (*)=mandatory field.
The grey box includes geometry tables which are not linked to record tables, but are used by the GUI to generate the final
geometry.