# Peer review of "Towards an operational European Drought Impacts Database (EDID)"

_EGUsphere, 2025_

## Author Response (AR1)

**Response to Editor,**

Thanks for inviting our revised manuscript for resubmission. Below we provide a detailed response to the comments by Reviewer 2 as requested. *The responses are in italic and blue font.*

**Kerstin Stahl**

**(on behalf of all coauthors)**

**Response to Reviewer 2**

This study presents the development of the European Drought Impacts Database (EDID), a significant step toward operationalizing drought impact monitoring within the Copernicus European Drought Observatory. The work addresses a critical gap in drought risk management by consolidating fragmented impact data and introducing a structured, operational framework. The paper is well-written, timely, and relevant for both scientific and operational communities.

Below, I provide general and specific comments aimed at improving clarity and completeness. Once addressed, I believe the manuscript will make a valuable contribution to NHESS.

General comments:

1) Scope of coverage: the manuscript states that EDID operates at the European scale; however, the dataset does not include all European countries and extends beyond the EU (e.g., Turkey). Please clarify the rationale for defining the geographic scope and selection criteria for included countries.

*Response: Funding came mostly from the EU, both for prior research projects and recently for the EDORA project that made possible the EDID development. Therefore, priority was given to EU countries as explained in the data section and, when data sources allowed, other countries were included too. We agree that the inclusion of databases from Turkey or Northern Africa will be highly desirable.*

*We added in line 418ff*

*"The inclusion of databases from further countries for example in Northern Africa or Turkey will be highly desirable, also considering that the physical drought indicators of the European Drought Observatory have been extended to that region. Hopefully such impact databases are being gathered in the respective countries for inclusion into EDID. This article aims to encourage such efforts and show the way to proceed."*

2) Long-term maintenance as a challenge / missing discussion point: While the Abstract and Conclusion mention EDID's operational integration within Copernicus, Section 4.2 does not explicitly discuss one of the main challenges: ensuring long-term maintenance and sustainability of the database under a specific institutional umbrella. Adding details on governance, funding, and community engagement plans would provide valuable insight into how EDID will remain operational and updated over time.

*Response: Indeed, long-term maintenance is a challenge. We added in line 422ff: ".*

*Maintenance and updates are common challenges for any operational monitoring system. For EDID they are ensured by the Joint Research Centre (JRC) in the framework of the Copernicus European Drought Observatory leveraging also its role as co-chair of the EU Working Group on Water Scarcity and Drought. JRC has already developed a long-term plan based on the Europe Media Monitor and the use of large language models and the engagement of drought experts that can contribute via a dedicated component of EDID by adding new records."*

3) Underrepresented systems: in Section 4.1 (around L306), the manuscript notes that some systems are less frequently represented. It would strengthen the discussion to explicitly name these systems and provide possible reasons for their lower representation (e.g., data availability, reporting practices, sector-specific challenges).

*Response: We agree to elaborate further and text was added in the revised version (line 321ff):*

"*The system ecosystems-terrestrial includes a large proportion of impacts on forestry and fewer impacts on a wide range of locally relevant other affected ecosystems. The drought response of trees often occurs with delay which might reduce awareness and reporting. It may be useful to combine the vegetation health indicators from remote sensing that are already used in many drought monitoring systems and the reported impacts, which can help the regional interpretation of these indices. The two energy systems and navigation are much less represented in EDID. One reason might be that they are more restricted to the specific locations where they can occur, i.e. where special infrastructure exists that can be affected (e.g. water transportation routes, hydropower plants). In addition, the impacts on people are less tangible and may be modified by market forces, which will likely affect the degree of awareness and limit reporting to regions where these impacts matter economically.*"

4) Additional challenges: Section 4.2 could be expanded to include other challenges, such as harmonizing impact reports with demographic or socio-economic factors (e.g., population density), which are critical for interpreting vulnerability and exposure.

*Response: Thanks for this important suggestion. The following text was added in 4.2, line 401ff:*

"*A consideration of cultural reasons why certain drought impacts are found worth reporting is largely lacking from data gathering, but will become important to consider when interpreting regional data. Demographics also control whether awareness propagates to an availability of reports and socioeconomic and political factors will affect the availability and quality of reports.*"

Specific comments:

Section 2.2 (L158): The text states that "measured and systematically observed data were found to be either weak or incomplete" and thus excluded from EDID. However, operational sources like propluvia were included. Please clarify why propluvia is not considered part of the "measured and systematically observed data" category to avoid confusion.

*Response: Thanks for pointing out that this is imprecise and indeed there is no very clear distinction. We revised the sentence to "More datasets were initially considered, but during the development of EDID many were found to be...".*

Abstract (L22): Consider adding a strong concluding sentence beyond "mobilizing the drought community," emphasizing the broader significance of EDID for risk management and policy.

*Response: Good idea - we added " The service provided by EDID therefore has the potential to contribute to drought risk management and policy for a drought resilient society".*

L28: Please specify what is meant by "risk" in contrast to "impact" (e.g., risk as potential impact) for clarity.

*Response: We would like to avoid a long discussion on where impacts fit into the risk concepts as there are different opinions about this and we don't want to restrict the interpretation or use of the data for any type of assessment. We therefore revised the sentence to read: "...do not directly provide data on observed impacts, nor on vulnerabilities that may contribute to drought risk."*

L30: "Impact-oriented" should start with lowercase "i."

*Response: Thanks, this was corrected.*

Figure 1 (L103): Indicate what the left and right screenshots represent for clarity.

*Response: Thanks, this was corrected and the caption now reads: " Screenshots of the European Drought Impact Database (EDID) showing starting screen and tabular options of basic impact counts by severity and system (left) and mapping of the locations where these impacts occurred (right), among other options to explore impact records through the Copernicus Emergency web application (https://drought.emergency.copernicus.eu/tumbo/edid)"*

Table 1: Typo: "Bachmaier" should be "Bachmair."

*Response: Thanks, this was corrected.*

Table 1: Please ensure all references listed in Table 1 appear in the Reference section (e.g., Bachmair et al., 2017 seems missing).

*Response: Thanks, errors were corrected.*

L199: Briefly explain what a Sankey diagram is for readers unfamiliar with the term.

*Response: Yes, we revised the text of the first bullet to " tracks the origin of the impact data and their classification into the systems. This is accomplished by visualizing the impact data flow from one categorization to another (with a "data flow" or "Sankey" diagram) ".*

Figure 2: Excellent synthesis of complex information. Two suggestions: 1) Consider improving readability by enlarging boxes or arranging text vertically. 2) Fig. would benefit from adding percentage values directly in the figure for "Original Source Type" and "EDID Systems" categories.

*Response: We tried if the numbers can fit in, but found it too difficult. Also, in general Sankey diagrams are not used with numbers. The graph becomes very full and slightly confusing when numbers are read horizontally (which one shouldn't as the % numbers are only valid for each column). We therefore prefer to leave it as it is, which is also the*

*convention and information is then not redundant to the text. We added to the caption that percentages for each column's categories are given in the text.*

Section 4.1 (L290): Consider revising the title for clarity: "4.1 A new baseline of drought impacts across Europe: spatio-temporal patterns."

*Response: Agreed, this was changed. Thanks for the suggestion.*